# Primary Antibiotic Resistance of *Helicobacter pylori* in Different Regions of China: A Systematic Review and Meta-Analysis

**DOI:** 10.3390/pathogens11070786

**Published:** 2022-07-12

**Authors:** Jinnan Chen, Puheng Li, Yu Huang, Yixian Guo, Zhaohui Ding, Hong Lu

**Affiliations:** 1Division of Gastroenterology and Hepatology, Shanghai Institute of Digestive Disease, Key Laboratory of Gastroenterology &Hepatology NHC Key Laboratory of Digestive Diseases, Renji Hospital, Shanghai Jiaotong University School of Medicine, Shanghai 200120, China; cjnbayern@sjtu.edu.cn (J.C.); 15626211482@163.com (Y.H.); devanxian@sjtu.edu.cn (Y.G.); dingzhaohui@renji.com (Z.D.); 2School of Mathematical Sciences, Peking University, Beijing 100091, China; lphleo@pku.edu.cn

**Keywords:** *Helicobacter pylori*, primary resistance, amoxicillin, clarithromycin, metronidazole, levofloxacin, tetracycline

## Abstract

Aim: Understanding the prevalence of antibiotic resistance can provide reliable information for selecting treatment options. The goal of this meta-analysis was to observe the primary antibiotic resistance of *Helicobacter pylori* (*H. pylori*) in different regions and time periods of China. Method: We searched PubMed, EMBASE, Chinese Biomedical databases and the China National Knowledge Infrastructure from inception to 20 February 2022. Data on the prevalence of *H. pylori* primary resistance at various time points were included. A random-effect model was established to calculate the pooled antibiotic resistance. Results: In total, 2150 articles were searched, with 70 meeting the inclusion criteria. The resistance to clarithromycin, metronidazole, levofloxacin amoxicillin, tetracycline and furazolidone in 2016–2020 were 34% (95% CI: 30–39%), 78% (95% CI: 73–84%), 35% (95% CI: 30–40%), 3% (95% CI: 1–5%), 2% (95%CI: 1–4%) and 1% (95% CI: 0–4%), respectively. Clarithromycin showed regional difference, as the resistance was higher in northern (37%, 95% CI: 32–41%) and western China (35%, 95% CI: 17–54%) than that in southern (24%, 95% CI: 17–32%) and eastern China (24%, 95% CI: 20–28%). Conclusion: The resistance of *H. pylori* to clarithromycin and metronidazole was high and increased over time, whereas resistance to levofloxacin, amoxicillin, tetracycline and furazolidone remained stable.

## 1. Introduction

Though decreasing in developed countries, the prevalence of *Helicobacter pylori* (*H. pylori*) is still high in China, causing a major health burden due to peptic ulcer disease complications and gastric cancer [1,2]. As an infectious disease, antibiotics-based therapies play a leading role in the treatment [3,4]. However, we face the serious challenge of high antimicrobial resistance because of the previous use of these antibiotics [5]. The primary antibiotic resistance decreases the efficiency of first-line treatment. The overall effect is dependent on both the cure rate with resistant strains and the proportion with resistance, especially clarithromycin and levofloxacin-containing regimen [6,7]. Empirical anti-*H. pylori* therapy is commonly used in current clinical practice instead of susceptibility-guided therapy which is unavailable in most of China. Therefore, obtaining high-quality local data and the antibiotic resistance pattern is needed to get good clinical outcomes [8]. In this study, we reviewed and analyzed primary antibiotic resistance rates of *H. pylori* in different regions and time periods in China over two decades to provide some guidance for selecting the first-line antibiotics.

## 2. Method

### 2.1. Search Strategy and Select Criteria

A search focused on the primary antibiotic resistance of *H. pylori* in the Chinese mainland population was done on Pubmed, Embase, the Chinese Biomedical (CBM) and the China National Knowledge Infrastructure (CNKI) from inception to 20 February 2022. The study was performed based on the Preferred Reporting Items for Systematic Reviews and Meta-Analyses (PRISMA) guidelines [9]. The search terms included were as follows: “Helicobacter pylori” and “China”, these search terms were combined with “Antibiotic resistance” and each individual antibiotic serially (“clarithromycin”, “metronidazole”, “levofloxacin”, “amoxicillin”, “furazolidone” and “tetracycline”).

In order to minimize selection bias, inclusion and exclusion criteria were established as follows: (1) the diagnosis of *H. pylori* infection must be based on at least one of the routine diagnostic methods (^13^C or ^14^C urea breath test, histology examination, rapid urease test or in vitro culture); (2) the patients had no use of Proton pump inhibitor, antibiotic or herbal medicine within the previous 2 weeks; (3) the patients did not receive *H. pylori* eradication therapy, which could exclude the influence of secondary drug resistance; (4) drug susceptibility was tested using the Agar dilution method, Epsilometer test (E-test), Polymerase Chain Reaction (PCR) or Kirby–Bauer (KB) disk diffusion method; (5) the patients were older than 18 years old; (6) the patients were mainland residents; (7) the articles must be original articles, not reviews or letters to editors. The inclusion of the article and data extraction were conducted by two authors. The disagreements were resolved by discussion between the two authors.

### 2.2. Data Extraction

Two authors (JN Chen and PH Li) extracted relevant information including: publication year, study period, source area, drug susceptibility method, number of patients enrolled and those with resistance of different antibiotics independently according to a standardized data extraction form. Yu Huang was responsible for the discordant results.

### 2.3. Statistics Analysis

Meta-analysis was performed for the primary antibiotic resistance of *H. pylori* in Chinese patients. In some studies, the *H. pylori* resistance rates were close to 0%. Therefore, Freeman–Tukey double arcsine transformation was used to process the data. Heterogeneity was tested by Cochran’s *Q* test and *I*² test (*I*^2^ < 25%, 25–75%, and *I*^2^ > 75% represents low, moderate and high heterogeneity, respectively). The DerSimonian and Laird random effect model was used to calculate the pooled rate and 95% confidence interval (CI). We used Egger’s test and Funnel plot to examine the potential heterogeneity. We calculated the mean annual percentage change and its confidence interval by calculating resistance rates between the earliest and most recent study periods available. For time period analysis, we divided the sample year into 4 groups based on their study period: before 2005, 2006–2010, 2011–2015 and 2016–2020. If the article spanned two time periods, we included it in its closest time periods. If the article included two or more time periods, we classified them separately. If the year of sample collection was not indicated in the study, 2 years before the publication of the article was defined as the study period [8]. For the region analysis, we divided China into four regions based on their geographical characteristics and matched each study, except the multicenter study, according to its urban location. All of the data analyses were performed using R version 4.1.0.

## 3. Results

We searched 2150 articles, and 70 articles were enrolled in the study (Figure 1). Among the included studies, 21 were from northern China, 27 were from eastern China, 8 were from southern China, 7 from western China and 7 studies were multicenter studies (Table 1).

### 3.1. Primary Resistance of H. pylori to Clarithromycin

The clarithromycin resistance sharply increased from 15% (95% CI: 9–22%) before 2005 to 34% (95% CI: 22–48%) in 2016–2020 (*p* < 0.001, Figure 2 and Appendix A). Meanwhile, the prevalence of clarithromycin resistance in different regions was also detected. The clarithromycin resistance in northern (37%, 95% CI: 32–41%) and western China (34%, 95% CI: 17–54%) were higher than that in eastern (24%, 95% CI: 20–28%) and southern China (24%, 95% CI: 17–32%) (*p* = 0.0004, Appendix A).

Subsequent time period analysis of northern and eastern China showed an upward trend of clarithromycin resistance in both regions over the past 20 years, from 16% to 42% (*p* < 0.0001, Figure 3 and Appendix A) and from 20% to 30% (*p* = 0.008, Figure 3 and Appendix A), respectively.

### 3.2. Primary Resistance of H. pylori to Metronidazole

The resistance rate of *H. pylori* to metronidazole steadily increased from 55% (95% CI: 44–65%) before 2005 to 78% (95% CI: 73–84%) in 2016–2020 (*p =* 0.0003, Figure 2 and Appendix A). Western China had the highest metronidazole resistance (83%, 95% CI: 65–95%), followed by eastern China (72%, 95% CI: 65–78%), southern China (68%, 95% CI: 54–82%) and northern China (64%, 95% CI: 60–68%). However, the difference did not reach statistical significance (*p =* 0.054, Appendix A).

Time period analysis was also performed. Despite the fact that no statistical difference was found in northern China (*p =* 0.279, Figure 3 and Appendix A), the metronidazole resistance showed a decreasing trend in the last 10 years, from 71% (95% CI: 63–78%) in 2006–2010 to 62% (95% CI: 57–67%) in 2016–2020. Similarly, no discrepancy was observed in eastern China within periods. However, the metronidazole resistance had increased from 63% (95% CI: 44–81%) to 79% (95% CI: 76–82%) numerically (*p =* 0.107, Figure 3 and Appendix A).

### 3.3. Primary Resistance of H. pylori to Levofloxacin

The levofloxacin resistance in China had decreased from 47% (95% CI: 35–58%) before 2005 to 24% (95% CI: 16%, 33%) in 2011–2015 but increased to 35% in 2016–2020 (95% CI: 30–40%) (*p =* 0.0186, Figure 2 and Appendix A). Regional variation had not been found with resistance estimates from northern (38%, 95% CI: 31–45%), eastern (32%, 95% CI: 27–37%), southern (21%, 95% CI: 10–36%) and western China (16%, 95% CI: 3–35%) (*p* = 0.087, Appendix A).

Further subgroup analysis showed there was no statistical difference of the levofloxacin resistance in northern China (*p =* 0.364, Figure 3 and Appendix A) and eastern China (*p =* 0.052, Figure 3 and Appendix A) during the same time periods. 

### 3.4. Primary Resistance of H. pylori to Amoxicillin, Tetracycline and Furazolidone

The primary resistance of *H. pylori* to amoxicillin (3%), tetracycline (2%) and furazolidone (1%) were low (Appendix A) and have remained relatively stable in the past two decades (Figure 3) 

### 3.5. Influence of Gender on the Primary Resistance of H. pylori to Clarithromycin, Levofloxacin and Metronidazole

The level of resistance depending on the gender of patients was shown in Appendix A. The results showed no difference in the resistance rates of clarithromycin (*p* = 0.5459), levofloxacin (*p* = 0.6522) or metronidazole (*p* = 0.2311) between male and female (Appendix A).

### 3.6. Meta-Regression Analysis of Antibiotics Resistance of H. pylori

Meta-regression analysis including regions, time periods and method was performed (Table 2) and indicated that compared with northern China and western China, eastern China (Difference: −0.23; 95% CI: −0.32, −0.14; *p <* 0.0.0001) and southern China (Difference: −0.17; 95% CI: −0.29, −0.05; *p* = 0.0066) had a lower risk of clarithromycin resistance. Southern China had the lowest levofloxacin resistance (Difference: −0.22; 95% CI: −0.40, −0.04; *p =* 0.0174). 

A positive correlation could be found between time periods and clarithromycin resistance based on multivariate analysis (*p <* 0.0001). Metronidazole resistance was higher in 2016–2020 (Difference: 0.22; 95% CI: 0.09, 0.35; *p =* 0.0009) than that before 2005. 

The method of the susceptibility test might also affect the results. Sub-analysis showed (Appendix A) the choice of susceptibility test method might affect the resistance rate of levofloxacin (*p* = 0.0013), amoxicillin (*p* < 0.0001), tetracycline (*p* < 0.0001) and furazolidone (*p* = 0.0293) rather than that of clarithromycin (*p* = 0.4019) and metronidazole (*p* = 0.0565). Further multivariate regression analysis demonstrated that compared with the agar dilution method, the disk diffusion method might overestimate the resistance rate of metronidazole (Difference: 0.13; 95% CI: 0.00, 0.25; *p =* 0.0417) and amoxicillin (Difference: 0.23, 95% CI: 0.15, 0.30; *p* < 0.0001) but underestimate that of levofloxacin (Difference: −0.21; 95%CI: −0.36, −0.06; *p* = 0.007). Meanwhile, E-test might overestimate the resistance rate of amoxicillin (Difference: 0.11; 95% CI: 0.01, 0.20; *p =* 0.0269) and tetracycline (Difference: 0.07; 95% CI: 0.01, 0.13; *p =* 0.0264) when compared with the agar dilution method.

## 4. Discussion

Supervising the prevalence of primary antibiotic resistance in a region can provide reliable information for the choice of treatment options [8]. 

In our study, we showed the mean overall resistance of *H. pylori* in China to clarithromycin, metronidazole and levofloxacin was 30.0%, 70.0% and 31.0% and increased over time, but that of amoxicillin, tetracycline and furazolidone was 3.0%, 3.0% and 1.0%, respectively, and remained low during these years. 

*H. pylori* has similar characteristics of clarithromycin and levofloxacin resistance with a clear mechanism by some certain gene mutation (23S rRNA and *gyrA*, respectively), which has an all-or-none effect on the efficacy. That means that the efficacy of treatment does not improve by increasing dose and duration [80,81,82]. Our study showed that the clarithromycin resistance in China has now reached 34%, while it seemed lower in eastern (24%) and southern China (24%). The levofloxacin resistance rate is currently 31%. Lower resistance could be found in the western China. In contrast, an increasing trend could be observed in eastern China, from 25% in 2011–2015 to 37% in 2016–2020. The resistance of clarithromycin and levofloxacin are both above the threshold of empirical use of these antibiotics.

Metronidazole, a class of nitroimidazole compound, is different from clarithromycin and levofloxacin, and the mechanism of resistance is not completely clarified at present. Meanwhile, different susceptibility methods or culture methods also affected the results as our previous work demonstrated that the resistance might be overestimated by E-test when compared with agar dilution in the area with high-level metronidazole resistance [83]. Previous studies have demonstrated that the resistance can be overcome by a high dose and long duration [80,84]. Our data showed that the resistance of metronidazole was 70%, ranging from 64 to 83% in different regions. It was noticeable that the metronidazole resistance in northern China had decreased from 71% to 62% in these years.

The overall primary resistance rate of *H. pylori* to amoxicillin, tetracycline and furazolidone remained low, and all of them were lower than 5%, with the exception of a few studies that reported higher rates of resistance.

Other studies from Asia, Europe and Latin America have also reported primary resistance of *H. pylori*, which was lower than that in China, as these data showed that clarithromycin resistance ranged from 12 to 21.4%, levofloxacin resistance ranged from 15 to 18% and metronidazole resistance ranged from 38.9 to 53% with an increasing trend over time.

The increasing resistance to clarithromycin, levofloxacin and metronidazole might be contributed to by the increasing consumption of these antibiotics and cross resistance to the corresponding antibiotics. Megraud et al. reported that the community consumption of these antibiotics was associated with its corresponding *H. pylori* resistance in European countries [5]. Similarly, Yang et al. found that the macrolide and quinolones ranked third and fourth in consumption of antibiotics, respectively, in China during 2018–2020 [85]. There are no definitive data on imidazole consumption in China. However, since metronidazole was produced in the 1960s, it had been widely used in the treatment of anaerobic infections in China. Compared with macrolide and quinolone, imidazole has been present in the community for a longer time, which has led to a high metronidazole resistance in China.

The success rate of clarithromycin-containing triple therapy has been reported less than 80% in China [86]. The addition of bismuth to the triple therapy, which has been recommended as a first-line therapy, improves cure rates despite a high prevalence of antimicrobial resistance. The effect of bismuth is to attain an additional 30–40% in the success with resistant infections [87]. The rising resistance to clarithromycin and levofloxacin can severely reduce the efficacy of this modified quadruple therapy failing to reach a 90% success rate. For metronidazole, resistance has no clinical significance since it can be overcome after increasing the dosage and prolonging the duration [80]. At present, amoxicillin is widely used in clinical practice, as long as patients have no allergic reaction. Despite the poor accessibility in China, tetracycline combined with metronidazole is often used as a first-line therapy in areas with high clarithromycin and levofloxacin resistance [3]. Furazolidone is a special drug, which is widely used in China due to its low resistance rate. Although it may be accompanied by adverse reactions such as peripheral neuritis, the treatment success rate is high [88].

There are still limitations in our review. Firstly, since it was a single rate meta-analysis, there was obvious heterogeneity among different studies. Second, most of the enrolled studies were from northern and eastern China and data from other regions was lacking, thus causing potential publication bias.

## Figures and Tables

**Figure 1 pathogens-11-00786-f001:**
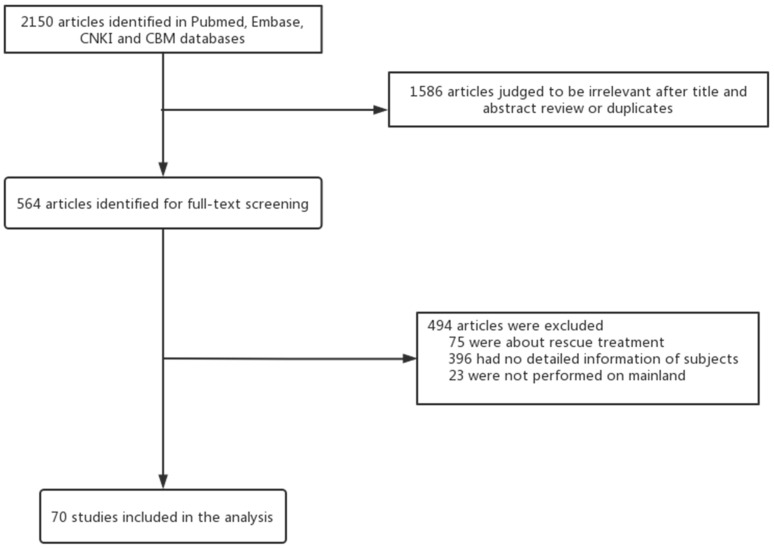
Study selection.

**Figure 2 pathogens-11-00786-f002:**
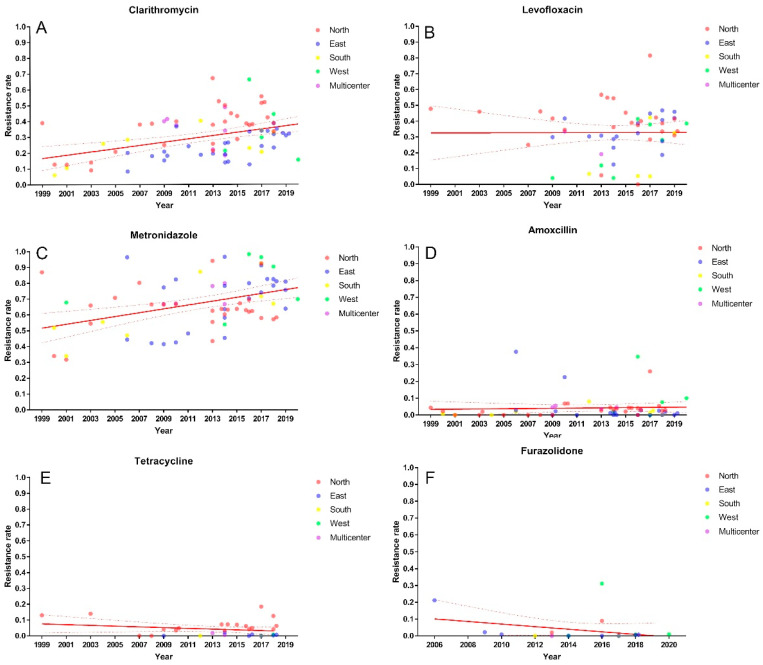
Primary clarithromycin (**A**), metronidazole (**B**), levofloxacin (**C**), amoxicillin (**D**), tetracycline (**E**) and furazolidone (**F**) resistance of *H. pylori* in China.

**Figure 3 pathogens-11-00786-f003:**
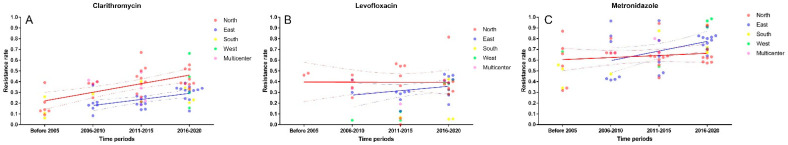
Time trends of primary clarithromycin (**A**), metronidazole (**B**) and levofloxacin (**C**) resistance in different regions of China.

**Table 1 pathogens-11-00786-t001:** Characteristics of the enrolled studies on resistance rate of *H. pylori* to antibiotics.

Authors	Regions	Year	Method	Clarithromycin		Metronidazole		Levofloxacin		Amoxicillin		Tetracycline		Furazolidone	
				Patients(n)	Prevalence(%)	Patients(n)	Prevalence(%)	Patients(n)	Prevalence(%)	Patients(n)	Prevalence(%)	Patients(n)	Prevalence(%)	Patients(n)	Prevalence(%)
**North China**														-	-
Gao et al. [10]	Beijing	2000	E-test	47	12.8	47	34.0	-	-	47	2.13	-	-	-	-
		2001		63	12.7	63	31.75	-	-	63	0.00	-	-	-	-
		2002–2003		22	9.09	22	54.55	-	-	22	0.00	-	-	-	-
		2004–2005		24	20.83	24	70.83	-	-	24	0.00	-	-	-	-
		2006–2007		71	38.03	71	80.28	40	25.00	71	0.00	41	0.00	-	-
		2008		39	38.46	39	66.67	39	46.15	39	0.00	39	0.00	-	-
		2009		24	25.00	24	66.67	24	41.67	24	0.00	24	4.17	-	-
Zhang [11]	Beijing	2009–2010	E-test	371	39.89	371	66.85	371	34.50	371	6.74	371	4.85	-	-
		2013–2014	E-test	950	52.63	950	63.37	950	54.84	950	4.42	950	7.26	-	-
Liu [12]	Beijing	2012–2013	PCR	130	37.69	-	-	-	-	-	-	-	-	-	-
Bai [13]	Beijing	2013	E-test	144	25.69	144	55.56	-	-	-	-	-	-	-	-
Zhang [14]	Beijing	2013–2014	E-test	700	50.14	700	63.86	700	54.43	700	3.71	700	7.29	-	-
Song [15]	Beijing	2013–2015	E-test	58	39.66	58	60.34	58	36.21	58	3.45	58	3.45	-	-
Li [16]	Beijing	2013–2020	E-test	74	51.35	74	58.11	74	28.38	-	-	-	-	-	-
Song [17]	Beijing	2014–2015	E-test	147	44.90	147	67.35	-	-	147	2.04	-	-	-	-
Suo [18]	Beijing	2014–2018	E-test	96	37.50	96	62.50	96	37.50	96	4.17	96	4.17	-	-
Suo [19]	Beijing	2014–2018	E-test	100	38.00	100	62.00	100	39.00	100	4.00	100	5.00	-	-
Fu [20]	Beijing	2015	E-test	324	43.21	324	63.89	324	45.37	324	4.32	324	7.10	-	-
Ma [21]	Beijing	2015–2016	E-test	56	28.57	56	69.64	56	0.00	56	0.00	-	-	56	8.93
Song [22]	Beijing	2015–2017	E-test	65	38.46	65	63.08	65	40.00	65	3.08	65	6.15	-	-
Fan [23]	Beijing	2015–2018	PCR	270	51.85	-	-	-	-	-	-	-	-	-	-
Song [24]	Beijing	2017–2018	E-test	650	33.54	650	58.46	650	33.69	650	2.92	650	4.15	-	-
Cui [25]	Beijing	2017–2018	E-test	506	38.74	506	57.31	506	31.03	506	2.57	506	6.32	-	-
Gao [26]	Beijing	-	PCR	111	42.34	-	-	111	41.44	111	5.41	111	12.61	-	-
Meng [27]	Hebei	2012–2013	KB	155	21.29	155	94.19	155	5.81	155	2.58	-	-	155	3.00
Wang [28]	Shandong	2011–2014	ADM	134	67.16	134	62.69	134	56.72	-	-	-	-	-	-
Wang [29]	Liaoning	1998–1999	E-test	23	39.13	23	86.96	23	47.83	23	4.35	23	13.04	-	-
		2002–2004	E-test	50	14.00	50	66.00	50	46.00	50	2.00	50	14.00	-	-
		2016–2017	E-test	27	55.56	27	92.59	27	81.48	27	25.93	27	18.52	-	-
Wang [30]	Shandong	2012–2014	ADM	-	-	101	43.56	-	-	-	-	-	-	-	-
**East China**															
Gu [31]	Shanghai	2005–2006	E-test	36	8.33	36	44.44	-	-	36	2.78	-	-	-	-
Lin [32]	Shanghai	2008–2009	KB	137	18.25	137	77.37	137	29.93	137	2.19	-	-	137	2.19
Zheng [33]	Shanghai	2008–2009	ADM	77	20.78	77	41.56	-	-	77	0.00	77	0.00	-	-
Sun [34]	Shanghai	-	ADM	133	18.05	133	42.11	-	-	-	-	-	-	-	-
Tan [35]	Shanghai	2009–2010	KB	120	36.67	120	82.50	120	41.67	120	22.50	-	-	120	0.83
Zhou [36]	Shanghai	2009	KB	248	15.32	248	42.74	-	-	-	-	-	-	-	-
Xu [37]	Shanghai	2010–2011	ADM	120	24.17	120	48.33	-	-	120	0.00	-	-	-	-
Liao [38]	Shanghai	2012	ADM	112	18.75	-	-	112	30.36	-	-	-	-	-	-
Hu [39]	Shanghai	2013–2015	E-test	132	14.39	132	63.64	132	30.30	132	0.00	-	-	-	-
Zhang [40]	Shanghai	2014	ADM	200	26.50	200	45.50	-	-	200	1.50	-	-	-	-
Shen [41]	Shanghai	2016	ADM	105	33.33	105	70.48	105	32.38	105	2.86	105	0.95	-	-
Long [42]	Shanghai	2016–2017	ADM	66	24.24	66	74.24	-	-	-	-	-	-	-	-
Chen [43]	Shanghai	2017–2018	ADM	382	35.08	382	82.72	382	46.86	-	-	-	-	-	-
Yu [44]	Shanghai	2018	ADM	145	31.72	145	81.38	145	40.69	145	0.00	-	-	-	-
Luo [45]	Shanghai	2018–2019	E-test	37	32.43	37	81.08	37	45.95	-	-	-	-	-	-
Xu [46]	Shanghai	2018–2019	ADM	100	32.00	100	64.00	-	-	100	0.00	-	-	-	-
Luo [47]	Shanghai	2018–2019	ADM	207	30.92	207	75.85	207	42.03	207	0.97	-	-	-	-
Cao [48]	Zhejiang	2005–2006	KB	85	20.00	85	96.47	-	-	85	37.65	-	-	85	21.18
Pan [49]	Zhejiang	2014	ADM	467	26.12	467	96.79	467	28.69	467	0.00	-	-	467	0.00
Liu [50]	Zhejiang	2016	ADM	398	12.56	398	80.15	398	38.69	398	0.00	398	0.00	398	0.00
Sun [51]	Zhejiang	2017	ADM	127	33.86	127	91.34	127	44.88	127	0.00	127	0.00	127	0.00
Xu [52]	Zhejiang	2018	ADM	56	23.21	-	-	-	-	-	-	-	-	-	-
Su [53]	Jiangsu	2013	PCR	159	19.50	-	-	159	30.82	-	-	-	-	-	-
Jiang [54]	Jiangsu	2017–2019	ADM	1204	38.62	1204	78.57	1204	27.41	1204	1.83	1204	0.33	1204	0.58
Jiang [55]	Jiangsu	2017–2019	KB	553	33.82	553	82.64	553	18.63	553	2.53	553	0.72	553	0.72
Liu [56]	Jiangxi	2010–2017	E-test	804	19.03	804	58.29	804	23.26	804	1.24	804	2.24	-	-
Hong [57]	Jiangxi	2014	E-test	374	13.90	374	78.36	374	12.57	-	-	-	-	-	-
**South China**															
Zhang [58]	Guangdong	2000	ADM	164	6.10	164	51.83	-	-	164	0.61	-	-	-	-
Yang [59]	Guangdong	2015–2017	PCR	244	22.95	-	-	244	5.33	-	-	-	-	-	-
Wang [60]	Guangdong	2016–2017	KB	39	20.51	39	71.79	39	5.13	39	2.56	-	-	-	-
Lu [61]	Guangdong	2016–2018	ADM	557	34.11	557	92.46	557	42.37	557	1.26	557	0.00	529	0.00
Zhang [62]	Guangdong	2017–2019	ADM	315	32.70	315	83.17	-	-	231	0.00	-	-	-	-
Ruan [63]	Fujian	2001	ADM	47	10.64	47	34.04	-	-	47	0.00	-	-	-	-
		2004	ADM	54	25.93	54	55.56	-	-	54	0.00	-	-	-	-
		2006	ADM	102	28.43	102	47.06	-	-	102	1.96	-	-	-	-
He [64]	Fujian	2019		-	-	-	-	262	32.44	-	-	-	-	-	-
Luo [65]	Guangxi	2011–2012	KB	300	40.33	300	87.33	300	6.67	300	8.00	300	0.00	300	0.00
**West China**															
Zhou [66]	Chongqing	2009	KB	-	-	-	-	100	4.00	-	-	-	-	-	-
		2013	KB	-	-	-	-	100	12.00	-	-	-	-	-	-
Yang [67]	Chongqing	2017	ADM	232	29.74	232	96.55	232	37.93	232	0.00	232	0.00	232	0.00
Zhou [68]	Chongqing	2012–2016	KB	150	21.33	150	54.00	150	4.00	150	2.00	-	-	52	0.00
Tang [69]	Sichuan	2017–2019	E-test	117	44.44	117	90.60	117	28.21	117	7.69	117	0.85	117	0.85
He [70]	Sichuan	2019–2020	KB	200	15.50	200	70.00	200	38.50	200	1.00	-	-	200	1.00
Hu [71]	Yunnan	2000–2001	E-test	-	-	109	67.89	-	-	-	-	-	-	-	-
Zhang [72]	Yunnan	2015–2016	KB	196	66.33	196	98.47	196	41.33	196	34.69	-	-	196	31.12
**Multicenter**															
Zhou [73]		2008–2010	E-test	280	40	280	66.79	-	-	280	4.64	-	-	-	-
Qi [74]		2008–2010	E-test	128	41.41	-	-	-	-	128	5.47	-	-	-	-
Song [75]		2008–2012	E-test	600	37.50	600	67.20	600	33.50	600	6.80	600	3.50	-	-
Xie [76]		2013–2014	E-test	288	18.40	-	-	-	-	288	4.51	288	0.69	-	-
Xie [77]		2013–2014	E-test	206	33.98	206	80.10	-	-	-	-	-	-	-	-
Zhou [78]		2013–2014	E-test	950	48.84	950	66.84	-	-	950	2.00	-	-	-	-
Liu [79]		2010–2016	E-test	1117	22.11	1117	78.25	1117	19.16	1117	3.40	1117	1.88	1117	0.00
Overall				18301	30.00	17013	70.00	14230	31.00	15448	3.00	10614	3.00	6045	1.00

**Table 2 pathogens-11-00786-t002:** Multivariate meta-analysis of antibiotics resistance of *H. pylori* in China.

	Clarithromycin		Metronidazole		Levofloxacin		Amoxicillin		Tetracycline		Furazolidone	
	Difference (95% CI)		Difference (95% CI)		Difference (95% CI)		Difference (95% CI)		Difference (95% CI)		Difference (95% CI)	
		*p* Value		*p* Value		*p* Value		*p* Value		*p* Value		*p* Value
**Period**												
Before 2005	Reference		Reference		Reference		Reference		Reference		NA	
2006–2010	0.22 (0.10 to 0.34)	0.0005	0.05 (−0.10 to 0.20)		−0.07 (−0.33 to 0.20)		0.07 (−0.03 to 0.16)		−0.19 (−0.31 to −0.07)	0.0019	Reference	
2011–2015	0.22 (0.11 to 0.33)	<0.0001	0.11 (−0.02 to 0.24)		−0.17 (−0.41 to 0.08)		−0.01 (−0.10 to 0.08)		−0.13 (−0.25 to −0.01)	0.0357	−0.18 (−0.46 to 0.11)	
2016–2020	0.28 (0.17 to 0.39)	<0.0001	0.22 (0.09 to 0.35)	0.0009	−0.04 (−0.29 to 0.21)		0.04 (−0.05 to 0.13)		−0.15 (−0.27 to −0.03)	0.012	−0.08 (−0.35 to 0.19)	
**Method**												
Agar dilution	Reference		Reference		Reference		Reference		Reference		Reference	
E-test	−0.08 (−0.18 to 0.02)		0.01 (−0.11 to 0.12)		−0.07 (−0.20 to 0.06)		0.11 (0.01 to 0.20)	0.0269	0.07 (0.01 to 0.13)	0.0264	0.09 (−0.21 to 0.39)	
Disk diffusion	−0.04 (−0.14 to 0.06)		0.13 (0.00 to 0.25)	0.0417	−0.21 (−0.36 to −0.06)	0.007	0.23 (0.15 to 0.30)	<0.0001	0.01 (−0.04 to 0.06)		0.12 (−0.08 to 0.32)	
PCR	−0.07 (−0.20 to 0.06)		NA		−0.14 (−0.34 to 0.06)		0.15 (−0.05 to 0.35)		0.20 (0.08 to 0.31)	0.0006	NA	
**Regions**												
North	Reference		Reference		Reference		Reference		Reference		Reference	
East	−0.23 (−0.32 to −0.14)	<0.0001	0.03 (−0.08 to 0.15)		−0.07 (−0.19 to 0.06)		−0.01 (−0.10 to 0.08)		−0.11 (−0.16 to −0.06)	<0.0001	−0.17 (−0.50 to 0.16)	
South	−0.17 (−0.29 to −0.05)	0.0066	0.03 (−0.14 to 0.19)		−0.22 (−0.40 to −0.04)	0.0174	0.00 (−0.11 to 0.11)		−0.15 (−0.22 to −0.08)	<0.0001	−0.20 (−0.54 to 0.13)	
West	−0.12 (−0.26 to 0.02)	0.0876	0.10 (−0.05 to 0.26)		−0.11 (−0.28 to 0.05)		0.05 (−0.06 to 0.16)		−0.13 (−0.20 to −0.07)	0.0001	−0.10 (−0.40 to 0.20)	
Multicenter	−0.05 (−0.15 to 0.07)		0.11 (−0.04 to 0.26)		−0.10 (−0.32 to 0.12)		0.02 (−0.06 to 0.11)		−0.08 (−0.13 to −0.03)	0.0032	−0.20 (−0.57 to 0.17)	

## Data Availability

Not applicable.

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
