# Peer review of "Primary Antibiotic Resistance of Helicobacter pylori in Different Regions of China: A Systematic Review and Meta-Analysis"

_pathogens, 2022, doi:10.3390/pathogens11070786_

Round 1

Reviewer 1 Report

The manuscript by Jinnan Chen et al. describes data on primary antibiotic resistance of H. pylori in different regions of China.

This review deserves to be thoroughly revised.

Line 46-47: should be in the methods section of the manuscript.

Line 67: who were the two authors? How were they selected? Are they the same authors who included the manuscript and extracted the relevant information? Why didn't the authors decide to include a third author for the discordant results?

Global: prefer passive voice.

Table 1. Bacteria name in italics. Because culture methods can have a significant impact, authors should discuss these parameters (e.g., for Metronidazole). Consider summarizing the data with a suitable figure.

Figures 1 and 2: rather than the resistance rate, prefer the plot of MIC versus year. Also, the resolution of the two figures is not sufficient.

Paragraph 3: beware of the hasty conclusion, when the CI95 intervals are superimposed (even partially) the difference cannot be considered as significant.

Disucssion: italicize bacterial gene names; "et al.

Authors' contribution: Did Puheng Li proofread the manuscript?

Reviewer 2 Report

Chen et al presented a meta-analysis investigated primary antibiotic resistance in China. The meta-analysis covers the whole period until 2020. This is a very well designed, prepared and written paper. Some modifications are needed.

Comments:

1.       At what age the patients are, it would be good the authors to extract the data and give the mean age and the range in Table 1 or group them depend age.

2.       If possible, the authors can analyze the level of resistance depending on the sex of the patients This information could be include in Table 1 or in separate table. The gender could influence resistance rates.

3.       Line 67 – For what period were these percentages? The phrase “present work” can be understood as the authors were presenting their own research. Please clarify.

4.       It will be interesting to read more about comparing the meta-analysis results with the H pylori resistance rates observed in countries in Asia, Europe, America.

5.       The authors can include data for the metronidazole consumption, the metronidazole resistance level is so high. It will be interesting how the authors can explain this fact.

Reviewer 3 Report

In the present systematic review with meta-analysis, Chen et al investigated prevalence and time trends of antibiotic resistance to H. pylori in China, in the period 2005-today. Main comments:

1) A meta-analysis should follow the PRISMA guidelines, which have not been cited.

2) Authors did not report how fixed or random effect model were chosen according to heterogeneity.

3) Since this is not a head-to-head comparison between groups, nut a cumulative pooled analysis for percentages, was inverse variance method used?

4) A sub-analysis of resistance rates according to diagnostic method is missing.

Author Response

Please see the attachment. we would appreciate your understanding.

Round 2

Reviewer 1 Report

Tha uthors have revised the manuscript according to my previous comments.

Author Response

Thanks a lot for your suggestion

Reviewer 3 Report

Regarding point 4, for "diagnostic method" I meant PCR vs E-test vs KB and so on.

Please answer about sub-analysis

Author Response

Point 1:   A sub-analysis of resistance rates according to diagnostic method is missing.

Thanks a lot for your kind suggestion. we have finished sub-analysis of susceptibility test method (Supplementary Figure 19-24) and showed in Page 9 Line 61-71

Round 3

Reviewer 3 Report

The paper may be accepted